# Dietary factors and predominant eye diseases in sub-Saharan African populations: A systematic review protocol

Isaiah Osei Duah Junior [1,¤a,☉]*, Kwadwo Owusu Akuffo [1], Josephine Ampong[1],
David Owiredu[2], Bridget Senya Boateng[3], Anthony Danso-Appiah[2,¤b,☉]

1 Department of Optometry and Visual Science, Kwame Nkrumah University of Science and Technology, Kumasi, Ghana, 2 Department of Epidemiology and Disease Control, School of Public Health, University of Ghana, Accra, Legon, Ghana, 3 Department of Clinical Medicine, Chengde Medical University, Chengde, Hebei, China

☉ These authors also contributed equally to this work.
¤aCurrent address: Department of Biological Sciences, Purdue University, West Lafayette, Indiana, United States of America
¤bCurrent address: Centre for Evidence Synthesis and Policy, School of Public Health, University of Ghana, Legon, Ghana
* oseiduahisaiah@gmail.com

## Abstract

Evidence linking diet and ocular diseases is growing, yet variations persist, with a paucity of data in sub-Saharan Africa. The proposed review will systematically synthesize evidence on dietary factors associated with predominant eye disorders (cataracts, refractive error, glaucoma, diabetic retinopathy, age-related macular degeneration, and dry eye disease) in the sub-Saharan African population. The systematic review protocol will follow PRISMA-P (Preferred Reporting Items for Systematic Reviews and Meta-Analysis Protocols) for transparency in reporting. All relevant published studies in the English Language will be identified from PubMed, Scopus, Web of Science, Embase, Health Inter-Network Access to Research Initiative (HINARI), Cumulative Index to Nursing and Allied Health Literature (CINAHL), African Journal of Science, and Cochrane Central Register of Controlled Trials (CENTRAL) using medical subject headings (MeSH) and controlled vocabulary without date restrictions. The reference lists of all retrieved studies will be checked and experts will be contacted for additional relevant studies. The risk of bias for observational studies will be assessed using ROBINS-E (Risk of Bias in Non-Randomized Studies - of Exposure) and for non-interventional and randomized studies ROBINS-V2 (Risk of Bias in Non-Randomized Studies version 2) and ROB2 (Cochrane Risk of Bias 2) will be employed respectively. Study quality will be assessed using the National Heart Lung and Blood Institutes Quality Assessment (NHLBI) tool for Observational Cohort and Cross-Sectional Studies and Controlled Interventional Studies. Meta-analysis will not be considered because of the wide range of dietary factors and the susceptibility

**Data availability statement:** This is a study protocol and no data has been generated thereof.

**Funding:** The author(s) received no specific funding for this work.

**Competing interests:** The authors have declared that no competing interests exist.

to high heterogeneity. Patterns of association between dietary factors and the specific eye diseases will be consolidated by Synthesis Without Meta-analysis (SWiM).

## Introduction

The burden of eye diseases and complications of blindness has increased exponentially and remains a public health threat globally [1] although it differs across geographical regions [2]. Uncorrected refractive error, cataracts, glaucoma, diabetic retinopathy, age-related macular degeneration, and dry eye disease remain high and are responsible for millions of cases of moderate to severe visual impairment and blindness [3–6]. Most cases occur in low-and-middle-income countries (LMICs) and territories [7,8]. In sub-Saharan Africa, for example, four million people live with blindness, and about 18 million have moderate-to-severe vision impairment [5]. The trends of ocular diseases within the region reflect a growing public health concern that necessitates urgent action.

In particular, cataract, characterized by cloudiness or opacity of the lens; uncorrected refractive errors, defined as the imbalance between axial length and refractive power of the eye; glaucoma, explained by progressive optic neuropathy with associated visual field changes; diabetic retinopathy, delineates the microvascular end-organ complication of diabetes mellitus; age-related macular degeneration expounds the deterioration of the macula and/or fovea; dry eye diseases described as the instability in tear film production, represents a major public health, economic, and societal challenges within this population [5,9-11]. For instance, in children, vision impairment from uncorrected refractive error has a negative impact on their academic prowess, self-image, and social relations [12–15]. Similarly, vision loss affects the career aspirations, employment prospects, and socioeconomic status of young adults and/or middle-aged individuals [16]. Likewise, in the older or aged population, derangement in vision culminates in physical dependency and psychological distress [17,18]. Most probably, a deficit in visual function reversibly affects the overall quality of life across the entire age spectrum [19–21].

Of note, diet is a modifiable factor involved in eye disease [22]. Specifically, nutritional factors, including food intake, dietary patterns, macronutrients, and micronutrients, are essential for optimal visual development [23–27]. The dietary nutrients possess biologically potentiating properties such as antioxidants [28,29], anti-inflammatory [29], anti-angiogenic [29], anticarcinogenic [30,31], neuroprotection [29] and light filtering properties [32,33], which are thought to confer protection against debilitating eye diseases. In particular, they mechanistically terminate the free radical reactions that result in oxidative stress, and ameliorate lipid peroxidation, which is hypothesized to mediate the pathobiology of several ocular diseases [34,35]. Given the underlying central dogma, we postulate that dietary factors could serve as an impetus to reduce the burden of eye diseases in sub-Saharan Africa as evidence of protective levels becomes available.

The evidence linking diet and ocular disease is controversial with no clear consensus. For example, there is still controversy about the adequacy [36,37] which limits its direct applicability in ocular health and vision medicine [25,38–40]. Similarly, the pattern of evidence from various systematic reviews and longitudinal studies are mixed [25,26,40–46]. Thus, while some authors report an association between higher intake and reduced risk of specific eye diseases [42,43], others report opposite [40,41,44,47], with most reviews showing no significant association [45,47,48] and a fewer more systematic reviews unable to draw a robust conclusion [25,26,29,39,45]. Despite the comprehensive nature of the above reviews, which provide invaluable insights, their focus of attention was not on the sub-Saharan population, which conceals its directedness.

Dietary diversity recognized as proxies for micronutrient sufficiency, quality, and adequacy of diets [49] are linked to socioeconomic status [50]. The differences in the standard of living in sub-Saharan Africa compared to the northern and southern African regions and the Westernized world differ in health status variables and nutritional indicators [51]. Corroborated by a multicenter cross-sectional study, Galbete and colleagues revealed dietary differences between sub-Saharan migrants and their local compatriots. While the latter group showed diverse eating patterns, the former category mainly consumed starchy staples and diets dominated by animal or animal-based products [51]. Acculturation, urbanization, and globalization are constantly aggravating the dietary habits of the inhabitants of the sub-Saharan Africa region, which means that an increase in the burden of the aforementioned ocular pathologies can be expected if concrete intervention strategies are not implemented. A systematic review is warranted to elucidate the dietary factors associated with prevalent eye diseases and, importantly, to propose sustainable nutritional intervention strategies for eye care in the sub-Saharan African region. The robustness, transparency, minimal bias, and inherent methodical rigor of this approach will bring to light the unknown, summarize the known, show masked trends, distilled insights, and provide a simple comprehensive regionally based recommendation for clinicians and policymakers. Further, the outcome of the study will transform nutritional policies geared towards reducing the burden of eye diseases in the region.

The overarching goal of the study is to systematically review the literature and synthesize the evidence on the dietary factors linked with the predominant ocular diseases in sub-Saharan Africa. Specifically, the review aims to answer the question, "What dietary factors are associated with predominant eye diseases (cataracts, refractive error, glaucoma, diabetic retinopathy, age-related macular degeneration, dry eye disease) in the sub-Saharan African population?" Specifically, the review will evaluate dietary intakes of foods (individual food items or broader food groups such as fruit or vegetables), macronutrients (carbohydrates, protein, fat, fibre), micronutrients (minerals, vitamins), dietary patterns (a posteriori or a priori) and their association with the individual eye diseases.

## Materials and methods

The current review protocol indexed in the International Prospective Register for Systematic Reviews (ID: CRD42023402042) has been prepared following the Preferred Reporting Items for Systematic Review and Meta-Analysis extension for protocols (PRISMA-P) checklist (S1 Table) [52,53] and the final review report will follow PRISMA for transparency of reporting [54,55] and the patterns of evidence for associations between dietary factors and eye diseases will be summarized according to Systematic Review Without Meta-Analysis (SWiM) [56].

### Inclusion criteria

The criteria for inclusion will follow the PICOS elements which comprises of the population, intervention, comparators, outcomes, and study design.

### Population

The study population will be all human participants of all ages living in sub-Saharan Africa who have been clinically diagnosed or self-reported to have either one or both eyes affected by either cataract, refractive error, glaucoma, diabetic retinopathy, aged-related macular degeneration, or dry eye disease in one or both eyes.

### Intervention

The interventions and/or exposures comprise dietary intakes of foods (individual food items or broader food groups such as fruit or vegetables), macronutrients (carbohydrates, protein, fat, fibre), micronutrients (minerals, vitamins), dietary supplements (as vitamins, minerals, lutein, zeaxanthin) or dietary patterns (a posteriori or a priori).

### Comparators

Comparators for the study where appropriate will be the control arm with the recommended dietary intake, dietary supplement, pharmacological and non-pharmacological interventions.

### Outcomes

Primary outcomes will be a reduction in prevalence, severity and/or progression of the specific eye diseases (cataract, uncorrected refractive error, glaucoma, diabetic retinopathy, age-related macular degeneration, and dry eye disease). The study will evaluate secondary endpoints including contrast sensitivity, visual acuity, pinhole acuity, glare, macular pigment optical density, color vision, night blindness, and retinitis pigmentosa.

### Study design

Eligible studies that will be considered for inclusion should provide empirical evidence on the linkage between dietary factors (excluding supplements) and eye diseases in human subjects in sub-Saharan Africa. Specifically, interventional studies, i.e., randomized and non-randomized control trials, and observational studies, such as cohort, case-control, and cross-sectional studies, investigating the effects and/or associations between dietary factors and the above-mentioned eye diseases will be included. Qualitative studies, case studies, case series will be included for exploratory purposes.

### Exclusion criteria

As the focus of the studies is on the sub-Saharan African population, studies investigating dietary factors and eye disease not within this population will not be considered for inclusion. Given our aim to synthesize only primary studies, systematic reviews, meta-analyses and overviews of reviews will be excluded. Commentaries, editorials, opinions, obituaries, animal studies, gray literature theses and/or capstones, preprints, and non-peer-reviewed articles will be excluded from the study due to their increased bias. Conference abstracts and articles without full text will be excluded, as they lack full details for critical evaluation and inclusion. In addition, animal studies, *in vitro* non-animal models, and *in silico* computational studies will be excluded from the study because they do not involve human subjects. The study will only include papers written in English, as most papers from this region use this language modality.

### Data sources, search terms, and search strategies for identifying studies

A comprehensive search will be performed in PubMed, Scopus, Web of Science, Embase, Health Inter-Network Access to Research Initiative (HINARI), Cumulative Index to Nursing and Allied Health Literature (CINAHL), African Journal of Science, Cochrane Central Register of Controlled Trials (CENTRAL) using medical subject headings (MeSH) and controlled vocabulary to identify relevant papers. Searches will be performed in English Language with no date restrictions and relevant Institutional databases. The search terms and strategy will be developed in consultation with a librarian to ensure robustness and replicability. The searches will be divided into three concepts (see Table 1 for search terms and search strategy). Adjustments will be made where appropriate to enable compatibility with the database. The Boolean operators "OR" and "AND" will be used for similar and different concepts respectively. Where appropriate, concepts will be adapted to optimize compatibility with the specific databases to enhance comprehensive retrieval of all relevant articles.

**Table 1. Search strategy developed to be adapted for the various databases.**

| Search | Concepts | Terms |
|---|---|---|
| 1 | dietary factors | diet[MeSH Terms] OR diet[Text Word] OR dietary pattern OR whole grain*OR refined grain*OR edible grain[MeSH Terms] OR Cereal[Text Word] *OR edible grain[MeSH Terms] OR grain[Text Word] OR grain* OR pasta*OR oryza[MeSH Terms] OR rice[Text Word] *OR potato*OR vegetables[MeSH Terms] OR vegetable[Text Word] *OR fruit [MeSH Terms] OR fruit[Text Word] *OR nuts[MeSH Terms] OR nut[Text Word] OR nut *OR fabaceae[MeSH Terms] OR legume[Text Word] OR legume* OR bean* OR ovum[MeSH Terms] OR egg[Text Word] OR egg* OR dairy OR dairies OR milk OR yogurt OR cheese OR fish OR seafood OR meat OR processed meat OR sugar sweetened beverage OR alcohol OR french fries OR pizza OR fast food OR tofu OR vitamins[MeSH Terms] OR vitamins[Text Word] OR vitamin* OR mineral* OR sugar* OR fat* OR lutein[MeSH Terms] OR lutein[Text Word]OR "zeaxanthins"[MeSH Terms] OR zeaxanthin[Text Word] |
| 2 | specific eye diseases | cataract[MeSH Terms] OR cataract[Text Word] OR glaucoma[MeSH Terms] OR glaucoma[Text Word] OR refractive errors[MeSH Terms] OR refractive error[Text Word] OR myopia[MeSH Terms] OR myopia[Text Word] OR hyperopia[MeSH Terms] OR Hyperopia [Text Word] OR astigmatism[MeSH Terms] OR astigmatism[Text Word] OR dry eye syndromes [MeSH Terms] OR dry eye syndrome[Text Word] OR macular degeneration [MeSH Terms] OR macular degeneration[Text Word] |
| 3 | Sub-Saharan Africa | sub-Saharan Africa OR Angola OR Benin OR Botswana OR Burkina Faso OR Burundi OR Cape Verde OR Cameroon OR Central African Republic OR Chad OR Comoros OR Democratic Republic of the Congo OR Djibouti OR Equatorial Guinea OR Eritrea OR Eswatini OR Ethiopia OR Gabon OR Gambia OR Ghana OR Guinea OR Guinea-Bissau OR Ivory Coast (Côte d'Ivoire) OR Kenya OR Lesotho OR Liberia OR Madagascar OR Malawi OR Mali OR Mauritania OR Mauritius OR Mozambique OR Namibia OR Niger OR Nigeria OR Rwanda OR São Tomé and Príncipe OR Senegal OR Seychelles OR Sierra Leone OR Somalia OR South Africa OR South Sudan OR Tanzania OR Togo OR Uganda OR Zambia OR Zimbabwe |
| 4 | Combine | Search: (#1) AND (#2) |
| 5 | Combine | Search: (#3) AND (#4) |

AND, Boolean logic to link different concepts; OR, Boolean logic to link the same concept.

## Screening and selection of studies

The retrieved articles from the respective databases will be consolidated into Endnote version 9 and imported into Covidence for duplicate removal and screening. A hierarchical three-dimensional approach will be employed in the screening process. Firstly, articles will be assessed based on titles and abstracts. Secondly, the full-text version will be critically evaluated for methodological quality and presence of effect sizes. Concurrently, following prespecified eligibility criteria, authors will include all feasible studies. Of note to eliminate bias and enhance transparency, all included papers will be separately and independently assessed by two authors. Where there is indecisiveness and/or where the lead researcher includes an article but one or more of the co-authors excludes, a consensus will be reached on that study by voting. Specifically, to agree on a decision a vote will be cast for inclusion or exclusion and in a 2:1 ratio in favor of a 'yes' or a 'no'. Conversely, if there is an equal proportion of votes on a decision, researchers will seek the opinion of an independent systematic reviewer before that study is either included or excluded. All methodological approaches and decisions that are taken prior to the selection of our analytical sample will be documented in a PRISMA flow diagram.

## Data extraction and management

The review will be conducted using Review Manager version 5.4. Data extraction will be performed on full-text articles by two independent authors using a pre-design data collection form. The data to be extracted will be categorized into study characteristics such as author name, year of publication, title, country, the primary focus of the review question, i.e., dietary factors and ocular diseases, review questions in terms of sample size, population, intervention, comparator, outcomes, and study design and/or primary findings on zero-order association between dietary factors and the specific eye diseases. Specifically, the most adjusted risk estimates, hazard ratios, risk ratios, or odds ratios for dichotomous

outcomes, and mean differences and standard deviations for continuous variables with their 95% confidence intervals will be extracted. Any disagreements will be resolved through discussion.

### Risk of bias

The risk of bias for observational studies will be assessed using ROBINS-E (Risk of Bias In Non-Randomized Studies - of Exposure), and that for non-interventional (Risk of Bias In Non-randomized Studies) and randomized trials ascertained using ROBINS-I and ROB2 (Cochrane Risk of Bias 2) respectively.

### Quality assessment

The quality of studies will be assessed by two independent authors using the National Institutes of Health Quality Assessment Tools for Observational Cohort and Cross-Sectional Studies and for Controlled Interventional Studies. Quality of the studies will be rated as poor, fair, and good.

### Data synthesis

The broader scope of the dietary factors will not permit pooled effect estimates by meta-analyses, therefore we plan not to perform heterogeneity, sensitivity, and meta-regression analyses. As statistically appropriate and to reach a meaningful conclusion, the findings will be consolidated narratively in accordance with Synthesis Without Meta-analysis (SWiM) [56]. Given the goal of the study to identify the role of dietary factors on the various eye diseases and the expected dietary differences between children and adults, we plan to synthesize data in a hierarchical manner. Specifically, we plan to systematically synthesize how each exposure affects the outcome, i.e., how each dietary factor affects each eye disease. In addition, we plan to isolate and uniquely highlight, where appropriate, dietary factors that affect the visual health of children.

### Patient and public involvement

Patients or the public were not involved in the design, conduct, reporting, or dissemination plans of our research.

### Discussion

Dietary factors play a key role in ocular health and diseases; including cataract [27], refractive error [57], diabetic retinopathy [25], dry eye disease [58], age-related macular degeneration [47], and glaucoma [26], given their putative antioxidant [59,60], anti-inflammatory[59,60], and role in metabolism [25,27,38,47,57,58,61]. Illumination of their beneficial and deleterious effects on eye health will be essential to mitigate the burden of these vision disorders and improve the quality of life of affected patients [24,62,63]. Yet, there remains a paucity of data in sub-Saharan Africa. The aim of the planned review is to systematically synthesize evidence on dietary factors associated with ocular diseases in sub-Saharan Africa. The findings from the review will provide regional-specific evidence and recommendations to inform ophthalmic clinicians and general physicians on the various dietary factors with beneficial and deleterious effects on the eyes which will invaluably help them advise their clients. Further, the results will guide policymakers to develop interventions to reduce the burden of eye disease in this part of the world. This review has several strengths and some limitations worth highlighting. For instance, this will be the first review to systematically synthesize evidence on the dietary factors that influence predominant eye diseases within the sub-Saharan African population. Second, the study will employ a more robust strategy that follows the PRISMA and SWiM guidelines where appropriate. Contrary, despite using multiple databases there remains a possibility of missing out on studies not indexed in these databases. However, this indexing bias is underscored by the review following a prespecified inclusion and exclusion criteria together with no date restrictions that extend the possibility of including all relevant studies to aid draw a robust conclusion.

## Supporting information

**S1 File. PRISMA-P (Preferred Reporting Items for Systematic Review and Meta-Analysis Protocols) 2015 checklist: Recommended items to address in a systematic review protocol.**
(DOCX)

## Acknowledgments

This systematic review protocol has been prepared as part of capacity building initiative by the Centre for Evidence Synthesis and Policy (CESP), University of Ghana, and Africa Communities of Evidence Synthesis and Translation (ACEST) that train experts in evidence synthesis and translation across low-income and middle-income countries (LMICs), particularly Africa. The lead author, Dr. Isaiah Osei Duah Junior, received mentorship from the Senior author, Professor Anthony Danso-Appiah; the Director, the Centre for Evidence Synthesis and Policy, University of Ghana, Accra). The authors appreciate the support of Dr. Moses Awuni (Department of Nutrition and Integrative Physiology, University of Utah, Salt Lake, UT, United States of America) and Dr Mary Adjepong (Department of Biochemistry and Biotechnology, Kwame Nkrumah University of Science and Technology, Kumasi, Ghana) for the helpful review and editing. The authors thank Dr. Jessica Eyeson (School of Medical Sciences, College of Allied Health Sciences, University of Cape Coast, Ghana) for proofreading the manuscript.

## Author contributions

**Conceptualization:** Isaiah Osei Duah Junior.

**Data curation:** Isaiah Osei Duah Junior, Kwadwo Owusu Akuffo, Anthony Danso-Appiah, Josephine Ampong, David Owiredu.

**Formal analysis:** Isaiah Osei Duah Junior, Kwadwo Owusu Akuffo, Anthony Danso-Appiah, Josephine Ampong, David Owiredu.

**Investigation:** Isaiah Osei Duah Junior, Kwadwo Owusu Akuffo, Anthony Danso-Appiah, Josephine Ampong, David Owiredu, Bridget Senya Boateng.

**Methodology:** Isaiah Osei Duah Junior, Kwadwo Owusu Akuffo, Anthony Danso-Appiah, Josephine Ampong, David Owiredu, Bridget Senya Boateng.

**Project administration:** Isaiah Osei Duah Junior, Kwadwo Owusu Akuffo, Anthony Danso-Appiah, Josephine Ampong, David Owiredu, Bridget Senya Boateng.

**Resources:** Isaiah Osei Duah Junior, Kwadwo Owusu Akuffo, Anthony Danso-Appiah, Josephine Ampong, David Owiredu, Bridget Senya Boateng.

**Software:** Bridget Senya Boateng.

**Supervision:** Anthony Danso-Appiah.

**Writing – original draft:** Isaiah Osei Duah Junior, Anthony Danso-Appiah, Bridget Senya Boateng.

**Writing – review & editing:** Isaiah Osei Duah Junior, Kwadwo Owusu Akuffo, Anthony Danso-Appiah, Josephine Ampong, Bridget Senya Boateng.

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
