## [Decision Letter · Decision Letter 0]

10 Mar 2025

PONE-D-25-00880Dietary Factors and Predominant Eye Diseases in Sub-Saharan African Populations: A Comprehensive Systematic Review ProtocolPLOS ONE

Dear Dr. Osei Duah Junior,

Thank you for submitting your manuscript to PLOS ONE. After careful consideration, we feel that it has merit but does not fully meet PLOS ONE’s publication criteria as it currently stands. Therefore, we invite you to submit a revised version of the manuscript that addresses the points raised during the review process.

We look forward to receiving your revised manuscript.

Kind regards,

Amin Sharifan

Academic Editor

PLOS ONE

Journal Requirements:

Additional Editor Comments:

1) As this is a protocol, please ensure all verb tenses are in the future tense.

2) Please note that PRISMA-P and other reporting guidelines are designed to improve transparency in reporting and are not 'gold standards' or guidelines for designing a study. Please revise your text accordingly. You may consult this reference for further information: 10.1186/s13643-021-01671-z.

3) There are two versions of the Cochrane risk of bias tool for randomized controlled trials. Please clearly indicate which version you will use for your analysis.

4) You have selected quite broad search platforms. I recommend omitting Google Scholar and BioMed Central from your search strategy. Google Scholar may not be an ideal platform for systematic reviews due to its broad scope and limited search functionality. Studies in BioMed Central should already be accessible through PubMed or Scopus.

5) Please consider using MeSH terms in your PubMed search.

6) I concur with the reviewers that the protocol appears rather ambitious, which could pose significant challenges during the execution of your work and potentially impede the production of a final output. In addition to the points raised by the reviewers, please consider that including both experimental and observational studies in your meta-analysis may result in substantial heterogeneity. Consequently, the findings may lack meaningful interpretation. I recommend focusing on a single study design if you intend to conduct a meta-analysis, or selecting previously mentioned study designs if you wish to proceed with a narrative synthesis of the results. For the latter, you may consider incorporating the SWiM reporting guideline in your protocol (10.1136/bmj.l6890), and the meta-analysis portion of the protocol would need to be omitted.

7) Given that you have already stated your intention to use a random effects model for meta-analysis, please articulate in the text why you would assess heterogeneity both between and within studies.

Reviewers' comments:

Reviewer's Responses to Questions

**Comments to the Author**

1. Does the manuscript provide a valid rationale for the proposed study, with clearly identified and justified research questions?

Reviewer #1: Partly

Reviewer #2: Partly

2. Is the protocol technically sound and planned in a manner that will lead to a meaningful outcome and allow testing the stated hypotheses?

Reviewer #1: Partly

Reviewer #2: Partly

3. Is the methodology feasible and described in sufficient detail to allow the work to be replicable?

Reviewer #1: No

Reviewer #2: Yes

4. Have the authors described where all data underlying the findings will be made available when the study is complete?

Reviewer #1: No

Reviewer #2: Yes

5. Is the manuscript presented in an intelligible fashion and written in standard English?

Reviewer #1: Yes

Reviewer #2: Yes

6. Review Comments to the Author

You may also provide optional suggestions and comments to authors that they might find helpful in planning their study.

Reviewer #1: This protocol describes a planned systematic review with meta-analysis of the relationship between dietary factors and a set of eye diseases, focusing on studies from sub-Saharan Africa. While the aim of this protocol is important, additional clarifications are needed before this protocol is ready for publication. I've recommended this manuscript for major revisions because I appreciate the time that will be required to consider and undertake the recommended revisions. I've organized my comments below into major and minor edits:

MAJOR:

1. Overall: This review is very broad, and the design risks combining effect sizes of different sizes and directions that are ultimately examining different underlying relationships, resulting in a series of statistically null pooled effect estimates but without clearly answering the research question. Additional clarification of the hypothesis would be helpful (see details below). If the aim is to identify any single factor or a small group of individual factors (e.g., antioxidants, as suggested in the Discussion) to improve ocular health, then such a broad approach is unnecessary; it would be better to instead focus on a few individual factors informed by the primary literature (and perhaps by pre-clinical studies). If instead the aim is to identify whether a diet characterized by healthier foods can reduce the risk of ocular disease, then it may be more useful to focus on dietary patterns or at least on a few food groups (e.g., fruits and vegetables, which are good sources of antioxidants).

2. Additional clarification is required to explain why another review on this topic is needed given the substantial number of other systematic reviews that show no statistically significant relationship or unclear conclusions and what we will learn from a new review that is specific to the region of sub-Saharan Africa. You clearly describe the higher prevalence of ocular diseases in this region and also the relatively lower diversity/quality of diets in this region. Please describe for readers how those 2 factors will result in a different/stronger conclusion in your proposed review. For example, is your hypothesis that these factors will give this review an improved signal-to-noise ratio, and thus greater ability to identify an effect?

3. As part of the clarification requested above, please shorten the Background to focus on the set-up to the review objective. At present, the included content is interesting, but some of it detracts from the main message.

4. Population: This section suggests that only participants with ocular diseases will be included, but the Outcomes section indicates that incidence rates will be considered. Please clarify either that the population are those at risk of ocular disease or the general population, or remove incidence from the outcomes.

5. Please describe how a medical librarian has/will contribute to the development of the search and/or peer review the search terms.

6. Data Analysis: Please describe how you will address evidence that cannot be meta-analyzed and which moderating variables you will examine. For the meta-regression, how do you propose grouping ages? Are there any other variable you will examine via subgroup analyses? Will all dietary factors be modeled together? It's not clear that all included factors listed are capturing the same biological pathway(s).

7. Measures of effect: Please describe how you will analyze the outcomes. Will you examine each of the above-mentioned diseases separately? How will you address the secondary outcomes? If so, that's a lot of models. If they are not truly independent, then p<0.05 may not be an appropriate cut-off for statistical significance.

MINOR:

1. Bridget Senya Boateng, Moses Awuni, and Dr Mary Adjepong listed in PROSPERO registration, but not here. If they have not met the criteria for authorship, please clarify their roles, using the Acknowledgement section if needed.

2. Background, Page 4, last paragraph: Please remove reference #47 from “with most reviews showing no significant association[46-49]” because it is not a review.

3. Intervention, Page 7, first paragraph, last sentence “to decrease either the prevalence, incidence, severity, and progression of the predominant eye diseases (cataract, uncorrected refractive errors, glaucoma, diabetic retinopathy, age-related macular degeneration, and dry eye disease).” Please remove this phrase, as it describes the outcome rather than the intervention.

4. Comparators section: “to mitigate the abovementioned eye diseases.” Please remove this phrase, as it describes the outcome rather than the comparator.

5. Study design: please add non-randomized trials and retrospective cohorts to either the list of inclusion or exclusion criteria, as appropriate.

6. Study design: Please remove the sentence “Data from national health examination surveys without clearly defined denominator, systematic reviews, opinions, commentaries will be excluded.” This is addressed below until “Exclusion criteria”

7. Please move the Header “Inclusion criteria” above “Population” to clarify that studies that include the above PICOS will be included. And please remove the first sentence currently under “Inclusion criteria”, as it is repeated above.

8. Exclusion criteria: What does “disproportionate or heterogeneous distributions” mean? Distributions of what?

9. Search terms: Will this search identify dietary patterns? This is not mentioned in the manuscript, but seems like a relevant dietary factor.

10. Screening and Selection of Studies: As written, it seems like all authors will independently screen each title, abstract, and full-text. Is that correct? 2 (or 3 if there is disagreement) seems appropriate.

11. ROBINS-I is appropriate for non-randomized interventions (see minor comment 5 if those will be included). For observational studies (cohort, case-control, and cross-sectional studies), please use ROBINS-E: https://www.riskofbias.info/welcome/robins-e-tool

12. Data Analysis: Please keep all descriptions of heterogeneity assessment together, either here in or below in a separate subsection.

13. Dealing with Missing Data: What does "incomplete" mean? If a study has an attrition rate >0%, will it be excluded?

14. Measures of Effect: Will you include Hazard Ratios, and how will you combine (or keep separate) the various ratios that will be reported?

15. Meta-Regression: This does not need to be repeated if included in the Data Analysis section.

16. Discussion: “Diet and dietary metabolites constitute the building…one-carbon metabolic cycle.” Please either directly link this to ocular function/disease risk or delete. As written, it is too general to be relevant.

Reviewer #2: Overall, I consider the topic to be relevant and of interest to the study population.

The most important issues I would like to highlight are firstly the overly broad research question. The list of nutrients (macro and micro) to be considered is too long, as well as the diseases to be researched. There are nutrients with known associations with visual health and related pathologies such as diabetes, hypertension, among others. The reason why these nutrients are chosen, or why they are all chosen, is not properly reflected in the background section. The fact that such a long list of nutrients has been chosen makes me doubt the feasibility of obtaining reliable, conclusive and concrete results.

On the other hand, in the case of finding studies in which, instead of measuring nutrients, they have given consumption as whole foods or dietary patterns, the source from which the nutritional composition data for certain foods would be obtained has not been specified.

The measures of outcomes, especially of severity or progression of the diseases described, are not specified.

The inclusion of the adverse effects study does not seem to be relevant to the research question.

In the exclusion criteria it is stated that articles in languages other than English will be excluded, yet in the search section and in the discussion it is stated that there will be no language restrictions. This is contradictory and needs to be corrected prior to publication.

The eligibility criteria for study designs are not sufficiently justified.

In the discussion there are no citations where appropriate.

The first reference in the reference list is incorrectly formatted, the author should be ‘Bourne R, et al.’.

7. PLOS authors have the option to publish the peer review history of their article (what does this mean? ). If published, this will include your full peer review and any attached files.

**Do you want your identity to be public for this peer review?** For information about this choice, including consent withdrawal, please see our Privacy Policy .

Reviewer #1: No

Reviewer #2: **Yes: ** Oana M. Craciun

---

## [Author Response · Author response to Decision Letter 1]

14 Mar 2025

Dr. Solna Carreon Santos

Editorial Manager

Plos One

PONE-D-25-00880R1

Dietary Factors and Predominant Eye Diseases in Sub-Saharan African Populations: A Comprehensive Systematic Review Protocol

We've checked your submission and before we can proceed, we need you to address the following issues:

1. Thank you for updating your data availability statement. You note that your data are available within the Supporting Information files, but no such files have been included with your submission. At this time we ask that you please upload your minimal data set as a Supporting Information file, or to a public repository such as Figshare or Dryad.

Please also ensure that when you upload your file you include separate captions for your supplementary files at the end of your manuscript.

As soon as you confirm the location of the data underlying your findings, we will be able to proceed with the review of your submission.

Thank you. This is a study protocol and no data has been generated thereof. We have indicated it in the revised manuscript.

---

## [Decision Letter · Decision Letter 1]

30 Mar 2025

PONE-D-25-00880R1Dietary Factors and Predominant Eye Diseases in Sub-Saharan African Populations: A Comprehensive Systematic Review ProtocolPLOS ONE

Dear Dr. Osei Duah Junior,

Thank you for submitting your manuscript to PLOS ONE. After careful consideration, we feel that it has merit but does not fully meet PLOS ONE’s publication criteria as it currently stands. Therefore, we invite you to submit a revised version of the manuscript that addresses the points raised during the review process.

We look forward to receiving your revised manuscript.

Kind regards,

Amin Sharifan

Academic Editor

PLOS ONE

Journal Requirements:

Additional Editor Comments:

1) Please note that PRISMA-P and other reporting guidelines are designed to improve transparency in reporting and are not guidelines for designing a study or preparing a work. Please revise both the abstract and main text accordingly. You may consult this reference for further information: 10.1186/s13643-021-01671-z.

2) Please present your revised search strategy as a supplementary file instead of the main text to enhance the flow and readability of your work.

3) Please justify the use of both risk of bias assessment and quality assessment. The former may suffice for a systematic review.

4) Please ensure the future tense is used throughout the text. The patient and public involvement section still contains past tense. This does not apply to the reporting or dissemination plans of your work as these will be done in the future.

5) Please ensure that your references are reported in accordance with the PLOS One style. Your may consult the followings for further information: https://journals.plos.org/plosone/s/submission-guidelines;
https://www.nlm.nih.gov/bsd/uniform_requirements.html.

Reviewers' comments:

Reviewer's Responses to Questions

**Comments to the Author**

1. Does the manuscript provide a valid rationale for the proposed study, with clearly identified and justified research questions?

Reviewer #1: Yes

Reviewer #2: Yes

2. Is the protocol technically sound and planned in a manner that will lead to a meaningful outcome and allow testing the stated hypotheses?

Reviewer #1: Partly

Reviewer #2: Yes

3. Is the methodology feasible and described in sufficient detail to allow the work to be replicable?

Reviewer #1: No

Reviewer #2: Yes

4. Have the authors described where all data underlying the findings will be made available when the study is complete?

Reviewer #1: Yes

Reviewer #2: Yes

5. Is the manuscript presented in an intelligible fashion and written in standard English?

Reviewer #1: Yes

Reviewer #2: No

6. Review Comments to the Author

You may also provide optional suggestions and comments to authors that they might find helpful in planning their study.

Reviewer #1: Thank you for addressing most of my comments from the first review. This protocol is close to being ready for publication, but a few more edits will ensure it meets reporting standards. Please see my recommendations below:

MAJOR:

1. The review is still very broad, which will make the evidence synthesis challenging and highlights the need for a clear synthesis plan. Because you intend to follow SWiM guidelines, please detail your synthesis plan in the Data Synthesis section, specifically including how studies will be grouped together for synthesis. Will any of the primary outcomes be combined or will you address them each separately? What about the exposures? Will you combine evidence for children and adults in a single summary statement? Will any other key variables be considered in the synthesis?

2. The protocol does not yet describe how a medical librarian has developed, or peer reviewed, the search. Please add that to ensure that the search will be thorough and replicable.

MINOR:

1. Thank you for adding clarifications about the objective and rationale in the Introduction. It would also be helpful to shorten to Introduction to focus on the objectives and rationale.

2. Line 116-117: both “micronutrients” and “dietary vitamins or minerals” are listed here. Please select 1.

Reviewer #2: The manuscript contains some grammatical errors that affect clarity. A thorough revision of the English language, preferably with the assistance of a native speaker or professional proofreader, is recommended.

Abstract (Main text line 27, and abstract on submission form) change "propose" to "proposed"

Main text line 80: change to "hypothesizes" or "which is hypothesized to"

Main text line 90: change "imply" to "implies"

Main text line 172: change "perform" to "performed"

7. PLOS authors have the option to publish the peer review history of their article (what does this mean? ). If published, this will include your full peer review and any attached files.

**Do you want your identity to be public for this peer review?** For information about this choice, including consent withdrawal, please see our Privacy Policy .

Reviewer #1: No

Reviewer #2: **Yes: ** Oana M. Craciun

---

## [Author Response · Author response to Decision Letter 2]

3 Apr 2025

PONE-D-25-00880R1

Dietary Factors and Predominant Eye Diseases in Sub-Saharan African Populations: A Comprehensive Systematic Review Protocol

PLOS ONE

Dear Dr. Osei Duah Junior,

Thank you for submitting your manuscript to PLOS ONE. After careful consideration, we feel that it has merit but does not fully meet PLOS ONE’s publication criteria as it currently stands. Therefore, we invite you to submit a revised version of the manuscript that addresses the points raised during the review process.

Thank you for the opportunity to resubmit the above-referenced manuscript. We have addressed all the reviewers’ comments, point-by-point, in our response below. The issues raised by the Reviewers are presented in normal font. We reply directly to these comments in bold font. All changes to the revised Manuscript have also been highlighted in red font.

Sincerely,

Dr. Duah

Thank you. We have addressed this query.

Additional Editor Comments:

1) Please note that PRISMA-P and other reporting guidelines are designed to improve transparency in reporting and are not guidelines for designing a study or preparing a work. Please revise both the abstract and main text accordingly. You may consult this reference for further information: 10.1186/s13643-021-01671-z.

Thank you for the suggestion. We have revised our statement as appropriate. Please see below and in the revised manuscript. Materials and methods. Page 6, Lines 118-123.

“The current review protocol indexed in the International Prospective Register for Systematic Reviews (ID: CRD42023402042) has been prepared following the Preferred Reporting Items for Systematic Review and Meta-Analysis extension for protocols (PRISMA-P) checklist (S1:Table 1) [56, 57] and the final review report will follow PRISMA for transparency of reporting [58, 59] and the patterns of evidence for associations between dietary factors and eye diseases will be summarized according to Systematic Review Without Meta-Analysis (SWiM) [60].”

2) Please present your revised search strategy as a supplementary file instead of the main text to enhance the flow and readability of your work.

Thank you. We have removed this section from the main text and showed in the supplementary file as suggested.

3) Please justify the use of both risk of bias assessment and quality assessment. The former may suffice for a systematic review.

Thank you for this opportunity. While risk of bias and quality assessment are often used interchangeably, there is a slight difference that we believe when both are combined as proposed in this review could improve the robustness of our evidence. For example, risk of bias aims to identify sources of systematic error, whereas quality assessment aims to determine the validity and reliability of the study thus the methodological soundness and rigour. Taken together, the quality assessment provides a holistic evaluation of the studies other than a single determination of systematic error.

4) Please ensure the future tense is used throughout the text. The patient and public involvement section still contains past tense. This does not apply to the reporting or dissemination plans of your work as these will be done in the future.

Thank you, we have extensively proofread the manuscript to reflect the tenses.

5) Please ensure that your references are reported in accordance with the PLOS One style. Your may consult the followings for further information: https://journals.plos.org/plosone/s/submission-guidelines;
https://www.nlm.nih.gov/bsd/uniform_requirements.html.

We have ensured that our referencing style is conforming to the PLOS referencing guidelines.

Reviewer #1: Thank you for addressing most of my comments from the first review. This protocol is close to being ready for publication, but a few more edits will ensure it meets reporting standards. Please see my recommendations below:

MAJOR:

1. The review is still very broad, which will make the evidence synthesis challenging and highlights the need for a clear synthesis plan. Because you intend to follow SWiM guidelines, please detail your synthesis plan in the Data Synthesis section, specifically including how studies will be grouped together for synthesis. Will any of the primary outcomes be combined or will you address them each separately? What about the exposures? Will you combine evidence for children and adults in a single summary statement? Will any other key variables be considered in the synthesis?

Thank you for the rigorous critical appraisal and though provoking questions which has helped us improved our work. We have addressed this query as seen below and in the Revised manuscript. Materials and Methods. Data Synthesis. Page 8, Lines 221-225

“Given the goal of the study to identify the role of dietary factors on the various eye diseases and the expected dietary differences between children and adults, we plan to synthesize data in a hierarchical manner. Specifically, we plan to systematically synthesize how each exposure affects the outcome, i.e., how each dietary factor affects each eye disease. In addition, we plan to isolate and uniquely highlight, where appropriate, dietary factors that affect children visual health.”

2. The protocol does not yet describe how a medical librarian has developed, or peer reviewed, the search. Please add that to ensure that the search will be thorough and replicable.

Thank you. We have provided a statement of such in the main text as requested. Materials and Methods. Data Sources, Search Terms, and Search Strategies for Identifying Studies. Page 8, Lines 172-173.

“The search terms and strategy will be developed in consultation with a librarian to ensure robustness and replicability.”

MINOR:

1. Thank you for adding clarifications about the objective and rationale in the Introduction. It would also be helpful to shorten to Introduction to focus on the objectives and rationale.

We have abridged the introduction to focus on the goals and rationale as suggested. However, we have retained some key details to provide context for the review.

2. Line 116-117: both “micronutrients” and “dietary vitamins or minerals” are listed here. Please select 1.

Thank you for pointing this out. Please we have deleted the “dietary vitamins or minerals” from the manuscript.

Reviewer #2: The manuscript contains some grammatical errors that affect clarity. A thorough revision of the English language, preferably with the assistance of a native speaker or professional proofreader, is recommended.

Thank you for your concern. We have done extensive proofreading of the manuscript and have enlisted the help of two other experts who have helped us improve the overall clarity and readability of the manuscript.

Abstract (Main text line 27, and abstract on submission form) change "propose" to "proposed"

We have incorporated the change. Please see below and in the Revised Marked Manuscript. Abstract. Page 2, Line 27.

“…proposed…”

Main text line 80: change to "hypothesizes" or "which is hypothesized to"

We have incorporated the change. Please see below and in the Revised Marked Manuscript. Introduction. Page 4, Line 80.

“…hypothesized…”

Main text line 90: change "imply" to "implies"

We have made the change. However, the line number is 95 instead of 90 as stated. See below and in the Revised Marked Manuscript. Introduction. Page 4, line 95.

“…implies…”

Main text line 172: change "perform" to "performed"

We have made the change. Please see below and in the Revised Marked Manuscript. Methods, Data Sources, Search Terms, and Search Strategies for Identifying Studies. Page 8, line 172.

---

## [Decision Letter · Decision Letter 2]

15 Apr 2025

Dietary Factors and Predominant Eye Diseases in Sub-Saharan African Populations: A Comprehensive Systematic Review Protocol

PONE-D-25-00880R2

Dear Dr. Osei Duah Junior,

We’re pleased to inform you that your manuscript has been judged scientifically suitable for publication and will be formally accepted for publication once it meets all outstanding technical requirements.

Kind regards,

Amin Sharifan

Academic Editor

PLOS ONE

Additional Editor Comments (optional):

1) Please ensure the title reflects only "systematic review." The term "comprehensive" is redundant because systematic reviews are inherently comprehensive by definition. For reference, see this article, which included 170 trials but appropriately used the title "systematic review". 10.1016/S0140-6736(22)00878-9.

2) Kindly update "PRISMA" to "PRISMA-P" in the abstract to align with the Preferred Reporting Items for Systematic Review and Meta-Analysis Protocols statement.

3) Page 7, line 150: Please remove the abbreviation RCTS as this is only used for randomized controlled trials and the correct form is RCTs.

Reviewers' comments:

Reviewer's Responses to Questions

**Comments to the Author**

1. Does the manuscript provide a valid rationale for the proposed study, with clearly identified and justified research questions?

Reviewer #1: Yes

Reviewer #2: Yes

2. Is the protocol technically sound and planned in a manner that will lead to a meaningful outcome and allow testing the stated hypotheses?

Reviewer #1: Yes

Reviewer #2: Yes

3. Is the methodology feasible and described in sufficient detail to allow the work to be replicable?

Reviewer #1: Yes

Reviewer #2: Yes

4. Have the authors described where all data underlying the findings will be made available when the study is complete?

Reviewer #1: Yes

Reviewer #2: Yes

5. Is the manuscript presented in an intelligible fashion and written in standard English?

Reviewer #1: Yes

Reviewer #2: Yes

6. Review Comments to the Author

You may also provide optional suggestions and comments to authors that they might find helpful in planning their study.

Reviewer #1: Nice work! Thank you for addressing my comments, both in the manuscript and in your response. This will be an important and interesting review, and I look forward to reading the final product. Good luck with the work ahead to complete this ambitious plan.

Reviewer #2: I believe it would be acceptable to be published at this stage.

Looking forward to the results of this systematic review.

7. PLOS authors have the option to publish the peer review history of their article (what does this mean? ). If published, this will include your full peer review and any attached files.

**Do you want your identity to be public for this peer review?** For information about this choice, including consent withdrawal, please see our Privacy Policy .

Reviewer #1: **Yes: ** Julie E.H. Nevins

Reviewer #2: **Yes: ** Oana M. Craciun

---

## [Editor Report · Acceptance letter]

PONE-D-25-00880R2

PLOS ONE

Dear Dr. Osei Duah Junior,

I'm pleased to inform you that your manuscript has been deemed suitable for publication in PLOS ONE. Congratulations! Your manuscript is now being handed over to our production team.

Kind regards,

on behalf of

Dr. Amin Sharifan

Academic Editor

PLOS ONE